# Catalytic and Electrochemical Properties of Ag Infiltrated Perovskite Coatings for Propene Deep Oxidation

**Thai Giang Truong [1,2], Benjamin Rotonnelli [1], Mathilde Rieu [3], Jean-Paul Viricelle [3], Ioanna Kalaitzidou [2], Daniel Marinha [1], Laurence Burel [2], Angel Caravaca [2], Philippe Vernoux [2,\* ] and Helena Kaper [1,\*]**

[1] Laboratoire de Synthèse et Fonctionnalisation des Céramiques, UMR 3080 CNRS/Saint-Gobain CREE, Saint-Gobain Research Provence, 550 avenue Alphonse Jauffret, 84300 Cavaillon, France; truongthaigiang@gmail.com (T.G.T.); benjamin.rotonnelli@ens-paris-saclay.fr (B.R.); danielmarinha@gmail.com (D.M.)

[2] Univ Lyon, Université Claude Bernard Lyon 1, CNRS, IRCELYON, F-69626 Villeurbanne, France; ioanna.kalaitzidou@ircelyon.univ-lyon1.fr (I.K.); laurence.burel@ircelyon.univ-lyon1.fr (L.B.); angel.caravaca@ircelyon.univ-lyon1.fr (A.C.)

[3] Centre SPIN, Mines Saint-Etienne Univ Lyon, CNRS, UMR 5307, 42023 Saint-Etienne, France; rieu@emse.fr (M.R.); viricelle@emse.fr (J.-P.V.)

\* Correspondence: philippe.vernoux@ircelyon.univ-lyon1.fr (P.V.); helena.kaper@saint-gobain.com (H.K.)

**Abstract:** This study reports the catalytic properties of Ag nanoparticles dispersed on mixed ionic and electronic conducting layers of LSCF ($La_{0.6}Sr_{0.4}Co_{0.2}Fe_{0.8}O_3$) for propene combustion. A commercial and a synthesized LSCF powder were deposited by screen-printing or spin-coating on dense yttria-stabilized zirconia (YSZ) substrates, an oxygen ion conductor. Equal loadings (50 μg) of Ag nanoparticles were dispersed via drop-casting on the LSCF layers. Electrochemical and catalytic properties have been investigated up to 300 °C with and without Ag in a propene/oxygen feed. The Ag nanoparticles do not influence the electrochemical reduction of oxygen, suggesting that the rate-determining step is the charge transfer at the triple phase boundaries YSZ/LSCF/gas. The anodic electrochemical performances correlate well with the catalytic activity for propene oxidation. This suggests that the diffusion of promoting oxygen ions from YSZ via LSCF grains can take place toward Ag nanoparticles and promote their catalytic activity. The best specific catalytic activity, achieved for a LSCF catalytic layer prepared by screen-printing from the commercial powder, is 800 times higher than that of a pure Ag screen-printed film.

**Keywords:** self-sustained electrochemical promotion; VOC abatement; silver catalyst; propene oxidation; mixed ionic electronic conductor

## 1. Introduction

PGM (Platinum Group Metals) based catalysts are commonly used for the abatement of pollutants, such as carbon monoxide and Volatile Organic Compounds (VOC) present in automotive gas exhausts and air cleaning [1]. However, there is a strong demand to substitute these costly and rare metals with equally effective and less expensive catalysts. Silver, around fifty-times cheaper than platinum, is a promising alternative. Ag-based catalytic coatings show a significant activity for CO and propene oxidation, especially under lean (oxygen excess) conditions, as encountered for air purification processes [2,3]. In this sense, one of the main tasks of these catalytic thin films is to optimize the surface/bulk ratio to save on raw materials while keeping a large exchange surface with the gas phase.

Catalytic thin films technology is particularly suitable for air cleaning where gaseous pollutants only "lick" the catalyst surface. In addition, catalytic films can be deposited on hot surfaces, such as collector walls in thermal engine exhausts for removing pollutants, or radiators for improving indoor air quality.

The aim of this study is to prepare efficient Ag-based catalytic coatings for propene deep oxidation. Propene is a representative alkene VOC emitted by thermal engines. This hydrocarbon is recognized as a pollutant due to its high photochemical ozone generation potential. Recently, we have shown that Ag catalytic layers deposited on yttria-stabilized zirconia (YSZ) dense membranes were catalytically active for propene combustion [3]. YSZ is an insulating ceramic that can maintain the heat produced by the exothermicity of the oxidation reaction on the catalytic coating, promoting hot points on catalytic active sites and enhancing therefore the catalytic performances. Furthermore, YSZ is an active support for catalysis due to its oxygen ion conductivity [4,5]. Oxygen anions contained in YSZ ($O^{2-}$) can act as promoting agents to modify electronic properties of the catalytic layer to achieve optimal catalytic activity. The impact of the oxygen ions supplied on the catalytic properties of metallic films was intensively investigated in the field of electrochemical promotion of catalysis (EPOC) [4]. The EPOC concept utilizes small electrical polarizations to control the oxygen ion migration from the YSZ solid electrolyte onto the catalytic layers. For instance, we have recently shown that the catalytic performance for propene oxidation of Ag films prepared by screen-printing can be enhanced by the electrochemical supply of oxygen promoting ions [3]. This beneficial effect was attributed to the production of more reactive oxygen species on Ag, due to the increase of the Ag work function and the concomitant weakening of the Ag–O bond strength. However, the development of efficient, robust and cheap reactors able to provide an easy polarization of catalytic films hinders the practical application of EPOC. Therefore, a research effort was recently focused on the possibility to achieve electrochemical promotion on metallic nanoparticles dispersed on ionic (YSZ) or MIEC (mixed ionic and electronic conductor) supports. This topic was named Self-Sustained Electrochemical Promotion (SSEP). A pioneer work was based on ethylene combustion on Rh nanoparticles dispersed on $TiO_2$, $WO_3$-doped $TiO_2$, $\gamma$-$Al_2O_3$, $SiO_2$, and YSZ, as well as on Rh continuous films interfaced on dense YSZ membranes [6]. Similar trends of the catalytic rate dependence with the oxygen partial pressure were observed either by varying the support work function (dispersed supported catalysts) or the applied potential. This demonstrates that a catalytic material can be promoted at the nanoscale by mobile ionic species coming from an oxide support without any external stimulus. A difference in work function between a metal and a conducting oxide could generate a driving force for the backspillover of oxygen ionic species from the support to the metal nanoparticles. One would expect that the magnitude of this driving force will depend on the ionic conductivity of the support, the temperature, and the catalyst/oxide interface at the nanoscale. For instance, this migration of oxygen ionic species from YSZ to Pt nanoparticles was at the origin of enhancement of the catalytic activity of Pt for propane combustion, as evidenced by $O_2$-temperature-programmed desorption [7] and isotopic oxygen exchange [8]. Similar conclusions were drawn from EPOC investigations, where positive polarization promoted the electrochemical migration of lattice $O^{2-}$ species toward the Pt surface, and increased the propane combustion rate of Pt films interfaced on YSZ dense membrane [4]. SSEP studies have recently been carried out on Pt, Rh, Ni, Ru, Ir, $Pt_{50}Sn_{50}$, and RuFe nanoparticles dispersed on YSZ and ceria-based supports for different kind of reactions such as ethylene oxidation, CO methanation, reverse water gas shift, $NO_x$ storage, and methane partial oxidation reforming [9–15].

Here, we aim to optimize the catalytic active phase by using more abundant Ag than PGM and by going from a continuous layer to (nano-sized) self-promoted particles. Electropromotion should occur via the transfer of oxygen ions from YSZ to a mixed electronic-ionic conductor. We choose LSCF ($La_{0.6}Sr_{0.4}Co_{0.2}Fe_{0.8}O_3$) for this study as this material shows already good conductivity at moderate temperature (300–400 °C), which is the temperature range for the here aspired application. However, LSCF reacts with YSZ at temperatures higher than 900 °C and forms insulating phases [16,17], and the two materials have different thermal expansion coefficients [18,19]. We, therefore, pay special attention to the preparation of the LSCF/YSZ layers by analyzing the initial powders, in particular with respect to

their grain size, and vary the deposition method (spin-coating and screen-printing). To the best of our knowledge, this study reports for the first time SSEP of Ag nanoparticles (NPs) dispersed on an MIEC porous layer supported on a YSZ disk for propene combustion. Ag NPs were dispersed in the porosity of LSCF layers, an MIEC oxide. To optimize the catalytic activity, we have prepared various LSCF layers with different thicknesses and porosities, using two distinct LSCF powders, one commercial ($LSCF_{com}$) and the other one prepared by the Pechini technique ($LSCF_{syn}$), showing different grain size distributions. LSCF layers were deposited on YSZ dense disks prepared by screen-printing and spin-coating. The preparation of these catalytic films is rather demanding: The LSCF layers need to adhere to the YSZ pellets, be homogeneous, crack-free and porous, in order to allow the infiltration with Ag NPs and interaction with the gas phase. Catalytic and electrochemical performances were investigated up to 300 °C for propene oxidation with and without polarization.

## 2. Results and Discussions

For this study, two different LSCF powders were employed, one synthesized using a modified Pechini route (see Section 3.1) and one commercial powder, denoted as $LSCF_{syn}$ and $LSCF_{com}$. In the first part, we analyze and compare the two bare powders, in particular with respect to their grain size distribution, in order to select adopted powders for the two employed deposition techniques, spin-coating and screen-printing (for detailed description of the protocols, see Section 3.2). In the following, we describe the preparation and characteristic of the LSCF/YSZ layers. And finally, we compare the catalytic and electrocatalytic performance of bare LSCF/YSZ samples with those of the Ag infiltrated LSCF/YSZ samples.

### 2.1. Characterization of Initial LSCF Powders

$LSFC_{com}$ shows a unimodal grain size distribution centered at around 1 μm (Figure S1), while $LSFC_{syn}$ has a trimodal distribution with maxima at 0.6, 14, and 50 μm. Despite the bottom–up approach to synthesize $LSCF_{syn}$, the obtained grains are relatively large. Since the grain size influences the preparation of the ceramic suspension, the layer thickness and the porosity of the films, $LSCF_{syn}$ was further modified by attrition milling. Indeed, after three minutes of attrition milling, the grain size of $LSCF_{syn}$ is decreased to a bimodal distribution with maxima at 0.4 and 3.4 μm (Figure S1). Prolonged milling of 30 min leads to even smaller grains (unimodal distribution around 0.3 μm), but at the same time increases the pollution of the materials with $ZrO_2$ traces from the milling balls. We therefore kept the milling time to 3 min, which is also close to the grain size of $LSCF_{com}$. SEM images of the different LSCF powders (Figure S2) are in line with the particle size distribution measurements from laser scattering (Figure S1). The three minutes' attrition milling ($LSCF_{syn}$_3min, Figure S2e,f) clearly decreases the particle size distribution.

XRD analysis (Figure S3) confirms the phase-purity of the different starting materials, all peaks can be indexed to rhombohedral LSCF (PDF 01-081-9113). Table S1 shows the results from X-ray fluorescence (XRF) analysis and $N_2$-physisorption. The chemical compositions of the different LSCF powders is in good agreement with the theoretical values. The specific surface area (SSA) of $LSCF_{syn}$ directly after synthesis is 7 $m^2$/g, and increases to 10 $m^2$/g after attrition milling. The specific surface area of $LSCF_{com}$ is slightly lower, 4 $m^2$/g (see Table S1). These values agree with the crystallite size determined using Scherrer's equation (Table S2). For $LSCF_{com}$, the crystallite size can be determined to 56 nm, while it is 26 and 34 nm for $LSCF_{syn}$ as-synthesized and after attrition milling, respectively.

### 2.2. Preparation and Characterization of LSCF Layers

$LSCF_{syn}$ and $LSCF_{com}$ were spin-coated on dense YSZ pellets (marked in the following as $LSCF_{syn}$–SC and $LSCF_{com}$–SC). The spin-coating protocol was adapted to both powder types in order to obtain homogeneous and mostly crack-free LSCF films (see Section 3, Materials and Methods, Table 1). The mass of deposited LSCF was kept constant to 2.7 ± 0.2 mg. The $LSCF_{com}$ loading obtained

by screen-printing (labelled as LSCF$_{com}$–SP) was higher (11 mg), as this technology is commonly used for the preparation of thick layers (see Table 2).

In order to have a good adhesion of the LSCF films on the YSZ pellets and avoid or limit the extensive formation of secondary phases at the oxide/YSZ interface [16,17], LSCF layers were calcined at 950 °C under static air for 4 h. The XRD patterns of the calcined LSCF/YSZ films are shown in Figure S4. The main crystalline phase is LSCF in the rhombohedral structure (PDF number 01-081-9113). Spin-coated samples include diffraction peaks assigned to YSZ support due the low thickness of the films. Small peaks attributed to traces of $SrZrO_3$ and $SrFeO_3$ can also be observed on the diffractograms of spin-coated layers. The presence of $SrZrO_3$ indicates that a chemical reaction at the interface between YSZ and LSCF took place during the calcination step at 950 °C of spin-coated samples. One can expect that a similar reaction occurs in the screen-printed layers, but it was not detected by XRD due to the thicker film.

Figure 1 shows the SEM images of the different LSCF layers deposited on YSZ. The morphology of the LSCF films prepared using LSCF$_{com}$ are rather similar, no major cracks are visible at the surface (Figure 1a,c) and the oxide films are fully covering the YSZ surface. Few macropores can be observed on the LSCF$_{com}$–SC layer, maybe due to a lower thickness compared to LSCF$_{com}$–SP. The LSCF$_{syn}$–SC film shows some surface cracks (Figure 1e). The smaller grain size of LSCF$_{syn}$–SC also leads to smaller pores (Figure 1f), leading to a broad distribution of pores throughout the sample. This layer prepared from LSCF$_{syn}$ is clearly more porous than the films prepared from the commercial perovskite.

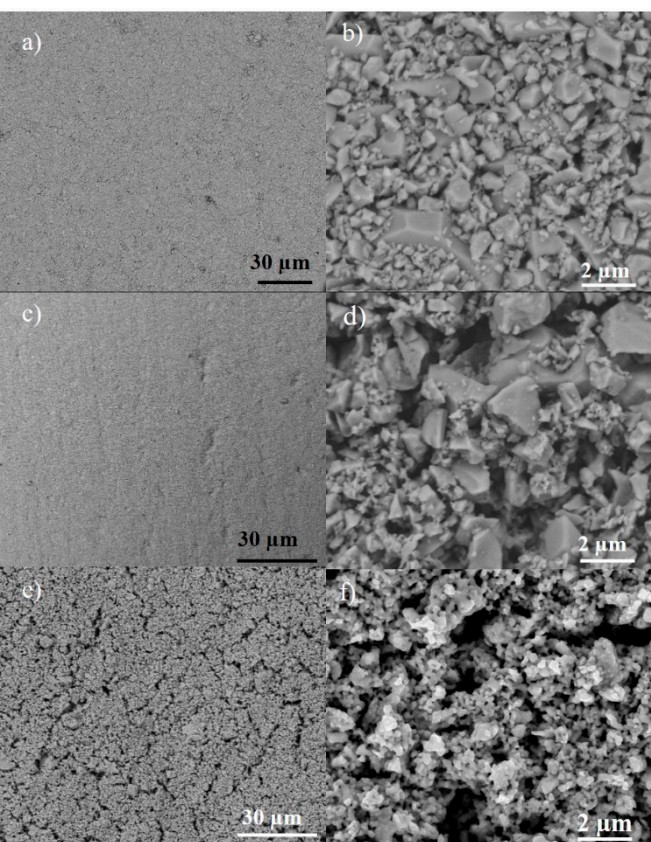

**Figure 1.** SEM images of the different LSCF ($La_{0.6}Sr_{0.4}Co_{0.2}Fe_{0.8}O_3$)/yttria-stabilized zirconia (YSZ) layers, taken at the center of the pellet. Low magnification (**a,c,e**) shows the surface state, while higher magnification (**b,d,f**) gives insight into the porosity of the layer. (**a**) and (**b**) LSCF$_{com}$–SP, (**c**) and (**d**) LSCF$_{com}$–SC, and (**e**) and (**f**) LSCF$_{syn}$–SC.

**Table 1.** Spin coating protocols for different LSCF films.

| Sample | Accelerating Rate (rpm/s) | Spinning Rate (rpm) | Spinning Time (s) |
|---|---|---|---|
| LSCFcom | 500 | 2500 | 10 |
| LSCFsyn | 500 | 2750 | 20 |

**Table 2.** Physical features of the LSCF/YSZ layers.

| Layers | Powder | Deposition Method | Thickness (μm) (Edge–Center) | Mass of LSCF (mg) | Mass of Ag (μg) |
|---|---|---|---|---|---|
| LSCF$_{com}$-SC | Commercial | Spin-Coating | 2–5 | 2.7 ± 0.2 | - |
| LSCF$_{syn}$-SC | Synthesized | Spin-Coating | 2–5 | 2.7 ± 0.2 | - |
| LSCF$_{com}$-SP | Commercial | Screen-Printing | 11 | 11 ± 1 | - |
| Ag/LSCF$_{com}$-SC | Synthesized | Spin-Coating | 2–5 | 2.7 ± 0.2 | 50 |
| Ag/LSCFsyn-SC | Synthesized | Spin-Coating | 2–5 | 2.7 ± 0.2 | 50 |
| Ag/LSCF$_{com}$-SP | Commercial | Screen-Printing | 11 | 11 ± 1 | 50 |

*2.3. Characterization of Ag-Infiltrated LSCF Layers*

Each LSCF film was impregnated with 50 μg Ag via drop-casting and was characterized by XRD after a reduction step at 300 °C used to decompose and reduce the Ag nitrate precursor (denoted as Ag/LSCF in the following). Diffraction peaks at 38.1°, 44.3°, and 64.4° (2θ°) corresponding to (1 1 1), (2 0 0) and (2 2 0) planes of fcc metallic silver prove the presence of surface metallic Ag (Figure 2). The presence of $Ag_2O$ could not be confirmed because of the overlap between the main peaks of $Ag_2O$ (1 1 1) (2θ = 33°) and those of LSCF, but cannot be excluded. The crystallite size of the Ag particles was estimated using the Scherrer's equation applied for the main diffraction peak of Ag at 38.2° (Table S2). Indeed, the crystallite size of Ag particles in the different LSCF layers is quite similar for all samples (ca. 50 nm for the two Ag/LSCF$_{com}$ layers and 46 nm for Ag/LSCF$_{syn}$).

Scanning electron microscopy was used to further determine the size of the infiltrated Ag particles, as well as their penetration into the whole LSCF layers along the cross section. SEM images of the surface layers evidence the presence of Ag particles (Figure 3) in the size range 50–400 nm (see also Figure S5 for higher magnification). The Ag/LSCF$_{com}$ film also contains micrometric Ag agglomerates (Figure 3a), suggesting an accumulation of Ag on the surface, predominantly at the centre of the layer. Ag particles are better dispersed on the surface on spin-coated films. Ag did not penetrate uniformly along the film thickness, despite the significant porosity observed on the bare LSCF layers. This is an indication of a possible low wettability of the layer by the Ag drops. Large micrometric Ag particles were not detected in Ag/LSCFsyn–SC (Figure 4e,f), in accordance with its higher porosity (Figure 1). In addition, TEM analysis (Figure 5) reveals the presence of Ag nanoparticles in the range 2–50 nm. These observations indicate that a wide size range of Ag particles, mainly located on the surface, exist in the three catalytic layers. Small Ag nanoparticles supported on LSCF are probably the catalytically most active ones.

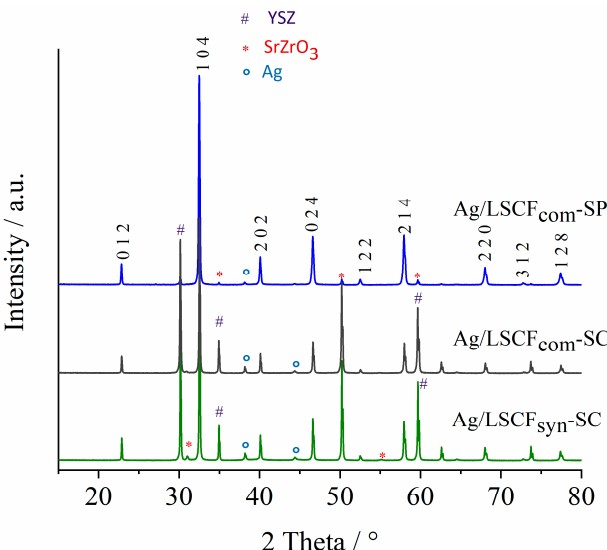

**Figure 2.** XRD of the Ag-infiltrated LSCF/YSZ films prepared by spin-coating (SC) and screen-printing (SP).

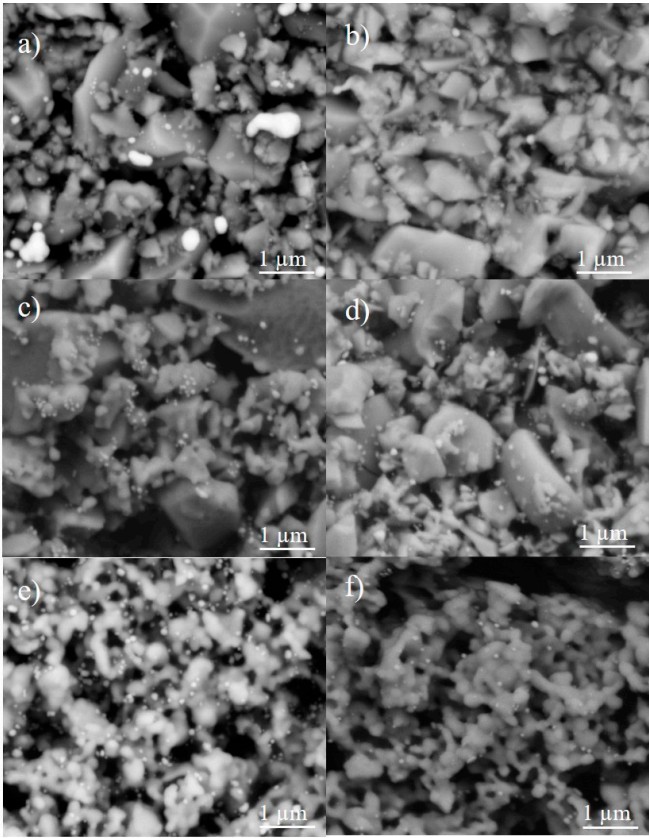

**Figure 3.** SEM images of the surface of Ag-infiltrated LSCF films after reduction at 300 °C for 2 h in 5% $H_2$/Ar. Images were taken at the center (left side, **a,c,e**) and edge (right side, **b,d,f**) of each film. (**a**) and (**b**) Ag/LSCF$_{com}$–SP (**c**) and (**d**) Ag/LSCF$_{com}$–SC, and (**e**) and (**f**) Ag/LSCF$_{syn}$–SC. Ag particles can be identified as brighter spots on the images.

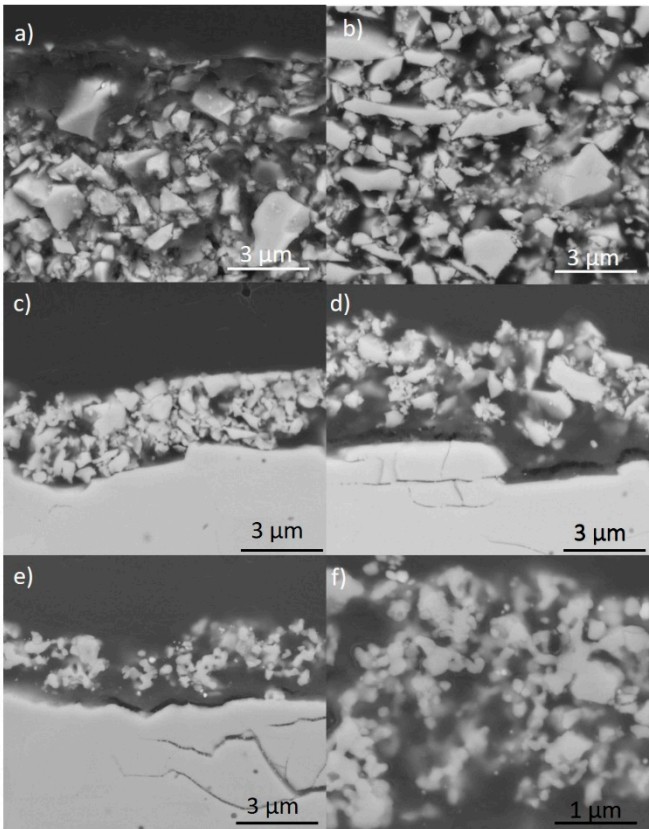

**Figure 4.** SEM images of the cross-sections of Ag-infiltrated LSCF films after reduction at 300 °C for 2 h in 5% H$_2$/Ar. Images were taken at the edge (left side, **a,c,e**) and center (right side, **b,d,f**) of each film. (**a**) and (**b**) Ag/LSCF$_{com}$–SP (**c**) and (**d**) Ag/LSCF$_{com}$–SC, and e) and f) Ag/LSCF$_{syn}$–SC. Ag particles can be identified as brighter spots on the images.

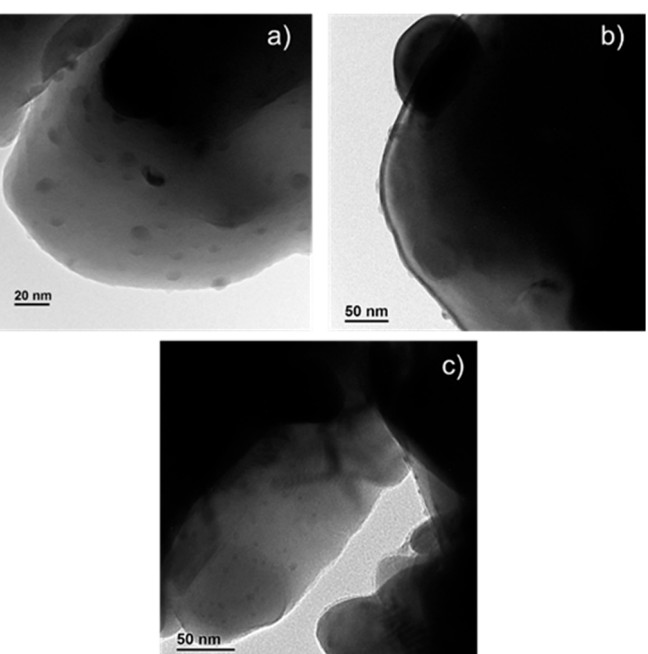

**Figure 5.** TEM images of (**a**) Ag/LSCF$_{com}$–SC, (**b**) Ag/LSCF$_{syn}$–SC, and (**c**) Ag/LSCF$_{com}$–SP.

*2.4. Catalytic Performances for Propene Oxidation*

2.4.1. Bare LSCF Coatings

Catalytic performances of the three LSCF layers for propene oxidation shown in Figure 6 are rather similar, despite the different mass of LSCF (see Table 2, m(LSCF) = 2.7 ± 0.2 mg for the spin-coated layers and 11 ± 1 mg for the screen-printed layers). The propene oxidation starts from around 200 °C and the conversion does not exceed 5% at 300 °C (Table 3), which are relatively poor performances. As expected, $LSCF_{com}$–SP shows a slightly better activity due to its higher mass of perovskite (Table 2). As the catalytic performances, the Open-Circuit Voltage (OCV) values at 300 °C are quite similar and positive, showing that the oxygen coverage on LSCF is high (Table 1). This indicates that the catalytic activity is limited by the propene chemisorption on the perovskite layers. Figure 7 displays cyclic voltammograms (20th cycle) at 300 °C in the propene/oxygen reactive mixture carried out between -2 V and +2 V with a scan rate of 10 mV/s. Except the first voltammogram (not shown in Figure 7), the following ones were identical for the three LSCF layers. As already observed for the catalytic properties, the three LSCF layers interfaced on YSZ show similar electrochemical performances. Cathodic performances are slightly better for layers containing $LSCF_{com}$, while the film deposited by screen-printing shows the best anodic performance. All in all, the $LSCF_{com}$–SP layer presents the best catalytic (Figure 6) and electrochemical (Figure 7) properties for propene oxidation. Recently, V. Tezyk et al. [20] have demonstrated that the shape of voltammetry curves depends on the LSCF layer microstructure. This way, the hysteresis and the intensity of anodic and cathodic peaks in the voltammogram can be linked to the surface oxygen exchange rate. For instance, in samples with high SSA, the oxygen exchange at the LSCF/gas interface is fast even at 300 °C. Therefore, the penetration depth of the electrochemical reaction in the LSCF layer is rather limited, leading to a weak hysteresis and small redox peaks. The $LSCF_{com}$-SP voltammetry curve shows the most pronounced hysteresis and the highest intensity of the cathodic peak in good agreement with its low porosity and high thickness. It is interesting to compare voltammograms of the two spin-coated layers which have similar thicknesses and oxide loadings. The slightly more intense cathodic peak of $LSCF_{com}$-SC indicates larger pores, in good agreement with the higher SSA of $LSCF_{syn}$ (Table S1). However, this small difference of porosity does not seem to have an effect on the catalytic performance (Figure 6).

**Table 3.** Catalytic performances and Open-Circuit Voltage (OCV) values of the bare and Ag-loaded samples.

| Catalytic Coatings | Propene Conversion at 300 °C/% | Specific Catalytic Rate at 300 °C /μmol $C_3H_6$ $s^{-1}.g^{-1}$ of Ag | OCV/mV |
|---|---|---|---|
| $LSCF_{syn}$-SC | 4 | - | +160 |
| $LSCF_{com}$-SC | 4.1 | - | +206 |
| $LSCF_{com}$-SP | 4.8 | - | +212 |
| Ag/$LSCF_{syn}$-SC | 14 | 95 | +303 |
| Ag/$LSCF_{com}$-SC | 17 | 115 | +220 |
| Ag/$LSCF_{com}$-SP | 33 | 225 | +251 |
| Sputtered Ag film [3] | 16 | 0.27 | +250 |

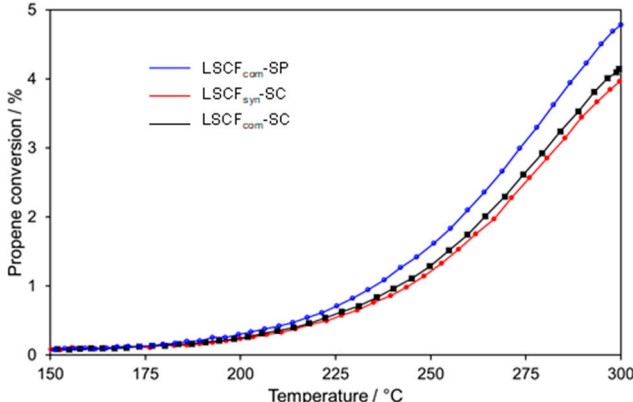

**Figure 6.** Catalytic performances for propene oxidation of bare LSCF layers. Test conditions: 0.1% $C_3H_6$ /3.6% $O_2$ (He carrier, total flow 3 L h$^{-1}$).

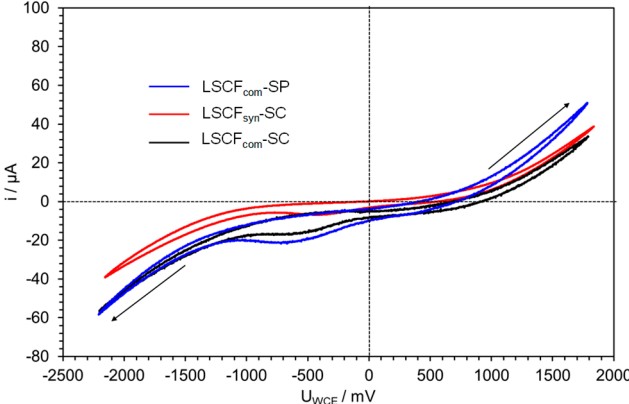

**Figure 7.** Comparison of the 20th cyclic voltammogram recorded at 300 °C in the reactive mixture on bare LSCF layers. The voltammograms are recorded between −2 V and +2 V with a scan rate of 10 mV/s.

### 2.4.2. Ag Infiltrated LSCF Catalytic Layers

The infiltration of 50 μg Ag in LSCF layers strongly improves the catalytic performances for propene oxidation (Figure 8). At 300 °C, the propene conversion was multiplied by 3.5, 4.1, and 6.9 for Ag/LSCF$_{syn}$–SC, Ag/LSCF$_{com}$–SC, and Ag/LSCF$_{com}$–SP, respectively. The catalytic performance of Ag/LSCF$_{com}$–SP is much higher than that of spin-coated layers, despite the similar Ag loadings. In addition, the most porous sample, Ag/LSCF$_{syn}$–SC, shows the lowest catalytic performance. Therefore, the catalytic activity of Ag infiltrated LSCF layers seems to be mainly linked to the presence of Ag on the top surface. This also suggests that the gaseous reactants, and most probably propene, cannot properly diffuse into the whole volume of the layers.

As expected, the presence of Ag does not significantly modify OCV values but improves the anodic electrochemical performances of LSCF$_{com}$-based films while the cathodic ones remain unchanged (Figure 9). Considering that Ag particles are mainly dispersed on the top surface of the layers, these results show that LSCF$_{com}$ particles act as an $O^{2-}$ pathway and allow the electrochemical oxidation of $O^{2-}$ onto Ag. On the opposite, Ag particles do not impact the electrochemical reduction of oxygen, suggesting that the rate-determining step is the charge transfer at the triple phase boundary (tpb) YSZ/LSCF/gas, where Ag is absent. Electrochemical anodic performances of Ag/LSCF$_{syn}$–SC are not significantly increased by the addition of Ag, indicating that the $O^{2-}$ diffusion into the LSCF$_{syn}$ layer is slower than into LSCF$_{com}$ based-films. This can be attributed to a higher porosity and/or a smaller size of LSCF$_{syn}$ grains.

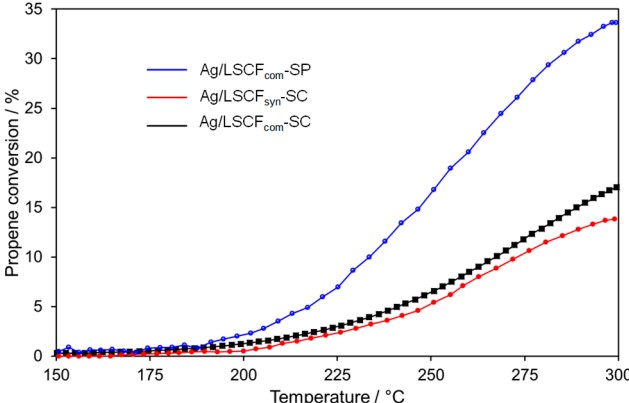

**Figure 8.** Catalytic performances of Ag-infiltrated LSCF layers. Test conditions: 0.1% $C_3H_6$/3.6% $O_2$ (He carrier, total flow 3 L $h^{-1}$). The silver particles were reduced in situ at 300 °C for 2h in 5% $H_2$/Ar (3 L $h^{-1}$) prior to the test.

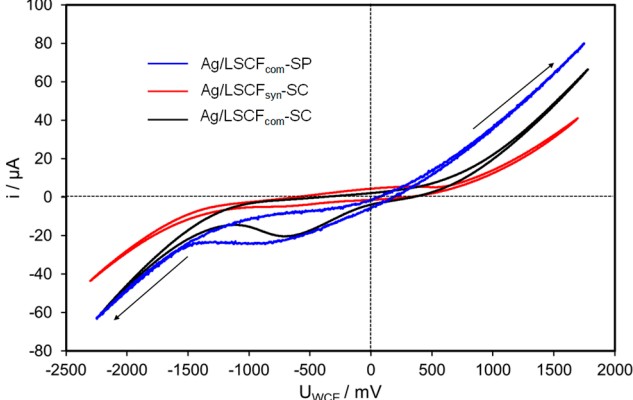

**Figure 9.** Comparison of the 20th cyclic voltammogram recorded at 300 °C in the reactive mixture on Ag-infiltrated LSCF layers. The voltammograms are recorded between −2 V and +2 V with a scan rate of 10 mV/s.

Figure 10a,b show the variation of the propene reaction rate at 300 °C as a function of the partial pressure of propene and oxygen for the most active sample, the Ag/LSCF$_{com}$–SP catalyst. The catalytic activity increases with the propene concentration, showing a positive order of around +0.7. The catalytic rate also increases with the oxygen partial pressure and reaches a plateau from approximately 2000 ppm of oxygen. These experiments suggest a low coverage of propene and oxygen on the surface at low oxygen concentrations. In the presence of an excess of oxygen (36,000 ppm) as in Figure 8, the activity seems to only depend on the partial pressure of propene (Figure 10a), indicating that the chemisorption of propene could be the rate-limiting step, in accordance with positive OCV values (Table 1). Different kinetic behaviours were reported in the literature on continuous pure Ag screen-printed films (without MIEC layer) interfaced on YSZ in the same experimental conditions at 300 °C [3]. Indeed, the catalytic activity shows a maximum with the propene concentration at 50–100 Pa of propene. The activity decreases upon further increase of propene concentration, suggesting a stronger chemisorption of propene on Ag compared to that of oxygen. Therefore, the electronic properties of Ag nanoparticles supported on LSCF are different to those of Ag micrometric grains contained in the screen-printed Ag films. Furthermore, the specific catalytic activity expressed per mass of Ag on Ag/LSCF$_{com}$–SP is around 800-times higher at 300 °C than that of a pure Ag screen-printed film (Table 1). This huge enhancement of the activity cannot only be explained owing to a higher Ag dispersion of Ag/LSCF$_{com}$-SP.

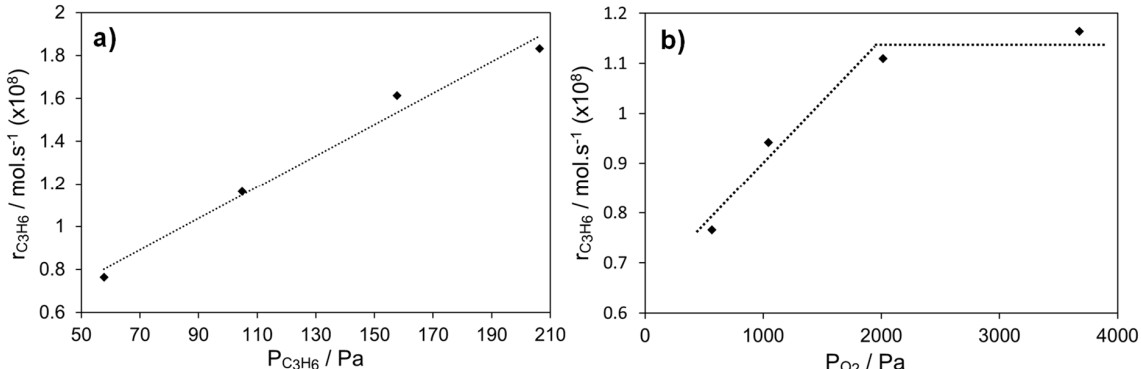

**Figure 10.** Catalytic rate on Ag/LSCF$_{com}$–SP versus (**a**) partial pressure of propene (PO$_2$ = 3600 Pa) and (**b**) partial pressure of oxygen (PC$_3$H$_6$ = 100 Pa). T = 300 °C.

The effect of small currents on the catalytic activity of the Ag/LSCF$_{com}$–SP layer was investigated at 300 °C (Figure S6). We did not observe any significant modification of the catalytic rate whatever the sense of the polarization. As discussed earlier, negative polarizations trigger the electrochemical reduction of oxygen at the tpb, where Ag nanoparticles are absent, or very rarely present. Therefore, the electronic properties of Ag nanoparticles were not modified by negative currents. On the other hand, voltammetry curves (Figure 9) have shown that the anodic performances are promoted by the presence of Ag in the LSCF$_{com}$-SP layer, showing that Ag nanoparticles are impacted by positive polarizations. According to the EPOC rules [4,21], positive polarizations increase the Ag work-function and then decrease the Ag electron density. Therefore, the chemisorptive bonds between Ag and propene, which is an electron donor, are strengthened, while Ag–O bonds are weakened as oxygen is an electron acceptor. This should result in the enhancement of the propene coverage and in the production of more reactive oxygen species. These two effects should be, according to the kinetic measurements (Figure 10), beneficial for the catalytic activity.

Such electrochemical promotion upon positive polarization of metallic nanoparticles dispersed in an MIEC layer has been reported in the literature on Pt and Pd nanoparticles for propane or methane oxidation [22–24]. However, the catalytic rate of Ag/LSCF$_{com}$–SP was found to slightly decrease under the application of the first positive polarization (+20 μA, Figure S6) for reaching a stable value not modified by further anodic polarizations. The non-promoting effect of positive polarizations may indicate that Ag nanoparticles dispersed on LSCF are already electropromoted, according to the SSEP concept. The higher the anodic electrochemical performances, the higher the catalytic activity. The diffusion of O$^{2-}$ promoting ions from YSZ via LSCF$_{com}$ grains promote the catalytic activity of Ag nanoparticles supported on LSCF. The catalytic performance of Ag/LSCF$_{com}$–SP was recorded at 300 °C for more than 15 h and was found to be relatively stable (Figure S6). As discussed earlier, the small decrease of the activity observed after around 6 h on stream was attributed to the positive polarization. TEM observations after this long-term catalytic test have confirmed that small Ag nanoparticles are still present on LSCF grains (Figure S7).

## 3. Materials and Methods

### 3.1. Preparation of LSCF Powders

The LSCF powder (designated in the following as LSCF$_{syn}$) was synthesized by a modified Pechini method [25,26] by using the following precursors: Fe(NO$_3$)$_3$·9H$_2$O (Alfa Aesar, 98%), La(NO$_3$)$_3$·6H$_2$O (Alfa Aesar, 99.9%), Sr(NO$_3$)$_2$ (Sigma Aldrich, >99%) and Co(NO$_3$)$_2$·6H$_2$O (Sigma Aldrich, > 99.99%). Typically, 17.810 g of Fe(NO$_3$)$_3$·9H$_2$O, 14.317 g of La(NO$_3$)$_3$·6H$_2$O, 4.665 g of Sr(NO$_3$)$_2$, 3.208 g of Co(NO$_3$)$_2$·6H$_2$O and 31 g of citric acid (CA) (Alfa Aesar, 99%) were dissolved in 140 mL water. The solution was stirred at room temperature for 1 h, followed by heating at 95 °C for 2 h, to reduce the volume to ca. 100 mL. At this point, 21 g of ethylene glycol (EG) was added. The reaction mixture

was further stirred for 4 h. The resulting brownish gel was calcined at 800 °C for 4 h (heating ramp: 3 °C/min). This latter was milled by attrition (Heidolph RZR 2102 Control mill) using a mixture of 160 g of isopropanol (Alfa Aesar), 1.5 kg of $ZrO_2$ balls (0.6 mm diameter) and 30 g of $LSCF_{syn}$. After removing the milling balls, the powder was dried at 100 °C overnight.

### 3.2. LSCF Deposition on YSZ Pellets

The dense YSZ substrates were synthesized from Yttria Stabilized Zirconia powder (TOSOH, 99.99%; average particle size 0.3μm), containing 8 mol% yttria, which was sintered at 1350 °C for 2 h (relative density higher than 98%). The final YSZ disks were 17 mm in diameter and 1 mm thick.

#### 3.2.1. Screen-Printing

A commercial LSCF powder with the same composition than $LSCF_{syn}$ (Fuel Cell Materials), denoted as $LSCF_{com}$, was used as received. Appropriate amount of $LSCF_{com}$ was mixed with a binder (ESL V400, 0.5 g per gram of powder) and a solvent (ESL T404, 8 drops per gram of powder) and homogenized by ball milling. The $LSCF_{com}$ layers were deposited on YSZ dense membranes using a semi-automatic screen-printer (METEOR from Reprint). After drying at room temperature overnight, the samples were calcined using the following protocol: 200 °C (6 °C/min, 1 h), 450 °C (0.5 °C/min, 2 h), 950 °C (2 °C/min, 4 h).

#### 3.2.2. Screen-Printing

Commercial and synthesized LSCF powders were deposited on dense YSZ pellets using spin coating from a ceramic suspension. Typically, 3 g of LSCF powder, 0.12 g of PEG (6000 M, Alpha Aesar), 0.53 g polyvinyl alcohol (900–10,000 M, 80% hydrolyzed, Sigma-Aldrich), 0.02 g contraspum (Zshimmer Schwarz) and 0.02 g Darvan CN (RT Vanderbilt) were dispersed in 3.26 g of distilled water. The prepared slurry was homogenously mixed by ball milling in a roll jar for 24 h and spin-coated onto the YSZ pellets. For each powder, the spin-coating protocol was adjusted in order to obtain homogeneous and crack-free layers after drying and calcination (see Table 1 for details). After drying at room temperature overnight, the samples were calcined using the same protocol as for screen-printed coatings.

A gold counter-electrode (CE) was deposited on the other side of the YSZ membrane disks by physical vapor deposition (magnetron sputtering, Cressingtion 208 HR). Gold was selected because of its negligible catalytic activity in propene oxidation, as checked via blank experiments under our experimental conditions.

#### 3.2.3. Ag Infiltration

LSCF layers were infiltrated with Ag using $AgNO_3$ as precursor via drop-casting. Since the LSCF layers prepared via spin-coating and screen-printing have different layer thicknesses (Table 2), we opted to keep the total amount of Ag deposited on the different films constant, i.e., 50 μg. From a freshly prepared stock solution containing 0.0787 g $AgNO_3$ and 5 g PEG in 0.5 l distilled $H_2O$, 0.5 mL were evenly distributed on the surface drop-wise. The freshly impregnated sample was dried at room temperature overnight. The list of catalytic layers and their main characteristics are reported in Table 2.

### 3.3. Characterization of LSCF, LSCF/YSZ, and Ag/LSCF/YSZ

XRD analysis were performed on a D8 Endeavor (Bruker) with $CuK_\alpha$ monochromatic radiation source (45 kV and 40 mA) and X'Celerator detector working in Bragg-Brentano geometry. Each diffractogram was measured between 5°–80° 2-theta with a step size of 0.01°. The grain size of the powders was determined using a HORIBA Laser Scattering Particle Size Distribution Analyzer LA-95. For each measurement, 10 mg of LSCF was dispersed in 50 mL of distilled water. The suspension was sonicated for 10 min. Measurements were carried out at 80% transmission waiting 0, 2, and 4 min after

sonication. $N_2$-physisorption isotherms were acquired on a Micromeritics Tristar II 3020 instrument at -196°C, after degassing the samples under vacuum at 200 °C for 12 h. The specific surface area (SSA) was calculated using the Brunauer–Emmet–Teller (BET) method. SEM measurements were carried out on a FEG FEI NOVA NanoSEM 230. The samples were prepared by drop casting the dispersed powder onto Cu/Zn supports. The bare LSFC films were metallized with Pt for 30 s (Cressingtion 208 HR). The Ag-infiltrated LSFC films were analyzed without metallization. All samples were observed ex-situ by Transmission Electronic Microscopy (TEM) before and after the catalytic tests using a JEOL 2010 microscope. The acceleration voltage was 200 kV with a 0.2 nm resolution. The catalyst coatings supported on the dense YSZ disks were scratched and dispersed in dry ethanol using an ultrasound bath. One drop of solution was then deposited on a copper grid for TEM measurements.

### 3.4. Catalytic and Electrochemical Measurements

Catalytic performances of the catalytic layers were performed in an experimental set-up described elsewhere [3,27]. The samples were placed in a continuous flow quartz reactor at atmospheric pressure. All catalytic tests were performed under a reaction mixture of 0.1% $C_3H_6$/3.6% $O_2$ (He carrier, 3 L h$^{-1}$). Prior to the catalytic experiments, the samples were heated up to 300 °C (2 °C/min) in 5% $H_2$/Ar (3 L h$^{-1}$) and reduced for 2 h at 300 °C. Then, the sample was purged with He (10 l h$^{-1}$) for 20 min prior to introducing the reaction mixture. The reactants and products were analyzed by an on-line micro gas-chromatograph (μGC-R3000, SRA equipped with two thermal conductivity detectors, a molecular sieve and a Porapak Q column for $O_2$, CO, $C_3H_6$, and $CO_2$ analysis) and an Infrared $CO_2$ analyzer (HORIBA VA-3000). Carbon monoxide was never detected, according to our 10 ppm lower detection limit. Reactants were Linde certified standards of $C_3H_6$ in He (8005 ppm), $O_2$ (99.999%) which could be further diluted in He (99.999%). The carbon balance closure was found to be within 2%.

The catalytic layer Working (W) and the Au counter electrode (CE) were connected to a potentiostat–galvanostat Voltalab PGZ402 (Radiometer Analytical). Open-Circuit Voltage (OCV) values were measured at 300 °C in the reactive mixture. Twenty successive cyclic voltammograms were recorded in the same conditions between −2 V and +2 V with a scan rate of 10 mV/s. The applied potential, ΔVWCE, was corrected from the OCV values.

## 4. Conclusions

In the present work, we study the catalytic and electrocatalytic performance of Ag infiltrated porous LSCF/YSZ layers for the catalytic oxidation on propene. The same loading (50 μg) of Ag nanoparticles was dispersed via drop-casting on LSCF layers, an MIEC oxide, prepared either by screen-printing or spin-coating from a commercial or synthesized powder. Ag nanoparticles are mainly located on the top surface of the catalytic layers. Electrochemical and catalytic properties have been investigated at 300 °C in a propene/oxygen feed with and without Ag. The presence of Ag nanoparticles does not influence electrochemical reduction of oxygen, suggesting that the rate-determining step is the charge transfer at the tpb YSZ/LSCF/gas, where Ag is absent. On the other hand, the anodic electrochemical performance correlates well with the catalytic activity for propene oxidation. This suggests that the diffusion of promoting oxygen ions from YSZ via LSCF grains can take place toward Ag nanoparticles and can promote their catalytic activity. The best specific catalytic activity, achieved for a LSCF catalytic layer prepared by screen-printing from the commercial powder, is 800-times higher than that of a pure Ag screen-printed film. This is the first example of Ag nanoparticles electropromoted via the SSEP concept and paves the path to further explore this concept of particular interest from an application point of view. Using SSEP for Ag nanoparticles allows replacing expensive platinum group metals by more abundant Ag. At the same time, the use of the catalytic active phase can be reduced by going from a continuous layer to dispersed self-promoted particles.

**Supplementary Materials:** The following are available online at http://www.mdpi.com/2073-4344/10/7/729/s1, Figure S1: Grain size distribution of LSCF$_{syn}$, LSCF$_{syn}$_3 min, LSCF$_{syn}$_30 min, and LSCF$_{com}$, Figure S2: SEM images of the different LSCF powders: a) and b) LSCF$_{com}$, c) and d) LSCF$_{syn}$, and e) and f) LSCF$_{syn}$_3 min.

Figure S3: XRD of LSCF$_{syn}$, LSCF$_{syn}$_3 min, LSCF$_{syn}$_30 min and LSCF$_{com}$. Figure S4: XRD of thin films of LSCF deposited on YSZ and calcined at 950 °C: LSCF$_{syn}$/YSZ, LSCF$_{syn}$_3 min/YSZ, and LSCF$_{com}$/YSZ. Figure S5: Impact of positive currents on the catalytic rate at 300 °C of Ag/LSCF$_{com}$–SP. Figure S6: TEM analysis of Ag/LSCF$_{com}$–SP film after catalytic testing, taken at the center of the LSCF layers. Table S1 XRF- and Surface area of LSCF$_{syn}$, LSCF$_{syn}$_3 min and LSCF$_{com}$ Table S2: Crystallite size of LSCF determined using Scherrer's equation (LSCF_com, LSCF_syn), LSCF deposited on YSZ and calcined at 950 °C (LSCF$_{com}$–SP, LSCF$_{com}$–SC, and LSCF$_{syn}$–SC) and Ag infiltrated into the LSCF layers (Ag/LSCF$_{com}$–SP, Ag/LSCF$_{com}$–SC, and Ag/LSCF$_{syn}$-SC).

**Author Contributions:** D.M., H.K., and P.V. designed the research plan and experimental outline, T.G.T. did the experimental work (powder synthesis, spin-coating, and characterizations), B.R. and T.G.T. optimized the spin-coating protocol, L.B. did the TEM analysis, M.R. and J.-P.V. carried out the screen-printing experiment, T.G.T., A.C., I.K., and P.V. carried out the electrochemical studies. H.K. and P.V. designed the manuscript with the help of all authors. All authors have read and agreed to the published version of the manuscript.

**Funding:** This study was performed within the "EPOX" project, funded by the French National Research Agency (ANR), ANR-2015-CE07-0026.

**Conflicts of Interest:** The authors declare no conflict of interest.

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
