# Peer review of "Catalytic and Electrochemical Properties of Ag Infiltrated Perovskite Coatings for Propene Deep Oxidation"

_catalysts, doi:10.3390/catal10070729_

Round 1
Reviewer 1 Report
Attempts to increase the activity of the La0.6Sr0.4Co0.2Fe0.8O3 catalyst in propene oxidation are interesting. Although the increase in perovskite activity when adding Ag nanoparticles is known.
My remarks:
- Why the studies shown only apply to a temperature of 300°C at which LSCF activity in propene oxidation is very low (propene conversion around 4%). Authors should show how catalyst activity changes at higher temperatures. If the conversion of propene on an LSCF catalyst does not increase significantly at higher temperatures, such a catalyst is not interesting.
- Maybe the LSCFcom-SC and LSCFsyn-SC layers used are too thin? It is obvious that with a thicker layer (e.g. LSCFcom-SP catalyst) conversion of propene increasing.
- The LSCFcom layers were deposited on YSZ and calcined at 950°C (2°C/min, 4 h). Why was the calcination temperature so high? The catalyst sintered significantly at this temperature, which reduced its activity.
- How did the authors determine that the Ag particle size is 50 nm (SEM Fig. 3) if the marker in the drawing is 1 mm?
- The chapter Catalytic and electrochemical measurements should specify: how much catalyst was used and what was the load on the catalyst.
- What does m LSCF and m Ag mean in Table 3? Are these LSCF and Ag masses?
- There are errors in the References. References 13,14,16,17, 20, 24 has no subscripts in the chemical formulas.
Author Response
We thank the reviewer for the comments. Below we wil answer the concerns/suggestions point by point.
- We study the catalyst activity below 300°C as this temperature range is of interest for the application, industrial VOc abatement. The conversion is indeed very low for the bare LSCF supports (ca. 4%). However, these measurements were carried out as blank tests for comparison with catalytic layers containing Ag. In presence of Ag, the conversion raises to 35% for Ag infiltrated LSFC layers (Fig. 8 in the manuscript). The conversion of 35% propene can be relevant and high enough upon recycling of the effluents over the catalyst bed.
- Indeed, the thickness of the layer might play a role in the catalytic performance. This is why we compare different techniques, spin-coating and screen-printing, which results in layers of different sizes. For more clarity, we added the thickness of the layers at the edge and centre in Table 3, which is actually different for the spin-coated samples.
- The choice of 950°C as calcination temperature was chosen to compromise between sintering/ evolution of secondary phases and good adherence of the LSFC layers on the YSZ pellets. Unfortunately, we could not avoid partial sintering of the LSFC layers.
- The Ag dispersion was also determined from images at higher magnification not shown in the current manuscript. We added images in the SI (Figure S5) to better demonstrate how the dispersion was determined. We also corrected our analysis for a particle size range between 50nm and 400 nm. We also like to draw the attention to the reviewer that the scale bare in Figure 3 is μm, not mm.
- These information have been added in the relevant sections and Figures.
- The referee is right, m in Table 3 is the mass of LSCF and Ag. We added this information in Table 3.
- We thank the referee for this comment. The references have been corrected.
Reviewer 2 Report
Interesting research, well prepared! I have a few comments to consider before posting results:
1) Introduction should be changed because it introduces little into the subject and does not highlight new products!
2)The part of the results is very unreadable (it combines preparation with analysis of results) I must admit that it is incomprehensible! Should be changed. - materials should be first!
3) Figure 1 - SEM a, c, e - what are they supposed to represent?
4) conclusions are to generall.
After the change, the work may be published in this Journal!
Author Response
We thank the reviewer for the comments. Below we answer the concerns/suggestions raised point by point.
- We added some aspects to the introduction. In particular, we added a paragraph on preparation of LSCF/YSZ layers.
Here, we aim to optimize the catalytic active phase by using more abundant Ag than PGM and by going from a continuous layer to (nano-sized) self-promoted particles. Electropromotion should occur via the transfer of oxygen ions from YSZ to a mixed electronic-ionic conductor. We choose LSCF (La0.6Sr0.4Co0.2Fe0.8O3) for this study as this material shows already good conductivity at moderate temperature (300-400 °C), which is the temperature range for the here aspired application. However, LSCF reacts with YSZ at temperatures higher than 900 °C and forms insulating phases,[16,17] and the two materials have different thermal expansion coefficients.[18,19] We, therefore, pay special attention to the preparation of the LSCF/YSZ layers by analyzing the initial powders, in particular with respect to their grain size, and vary the deposition method (spin-coating and screen-printing).
- The structure of the manuscript follows the template of Catalysts, where the materials section is placed after the Results and Discussion part. We agree with the reviewer that therefore the Results and Discussion part needs an introduction on the strategy and sample. We added the following paragraph:
For this study, two different LSCF powders were employed, one synthesized using a modified Pechini route (see section 3.1) and one commercial powder, denoted as LSCFsyn and LSCFcom. In the first part, we analyze and compare the two bare powders, in particular with respect to their grain size distribution, in order to select adopted powders for the two employed deposition techniques, spin-coating and screen-printing (for a detailed description of the protocols, see section 3.2). In the following, we describe the preparation and characteristic of the LSCF/YSZ layers. And finally, we compare the catalytic and electrocatalytic performance of bare LSCF/YSZ samples with those of the Ag infiltrated LSCF/YSZ samples.
3. We thank the reviewer for this comment. In Fig. 1 we compare the surface of the different LSCF/YSZ layers at two different magnifications, in order to demonstrate that the obtained films are basically crack-free and porous. We changed the title of this Figure for more clarity:
Figure 1. SEM images of the different LSCF/YSZ layers, taken at the center of the pellet. Low magnification (a, c, e) shows the surface state, while higher magnification (b, d, f) gives insight into the porosity of the layer. a) and b) LSCFcom-SP, c) and d) LSCFcom-SC and e) and f) LSCFsyn-SC.
4.
Our conclusion was focused on the main results of this paper. We have modified the conclusion to remind the objective of the study and to highlight its novelty
In the present work, we study the catalytic and electrocatalytic performance of Ag infiltrated porous LSCF/YSZ layers for the catalytic oxidation on propene. The same loading (50 µg) of Ag nanoparticles was dispersed via drop-casting on LSCF layers, a MIEC oxide, prepared either by screen-printing or spin-coating from a commercial or synthesized powder. Ag nanoparticles are mainly located on the top surface of the catalytic layers. Electrochemical and catalytic properties have been investigated at 300 °C in a propene/oxygen feed with and without Ag. The presence of Ag nanoparticles does not influence electrochemical reduction of oxygen, suggesting that the rate-determining step is the charge transfer at the tpb YSZ/LSCF/gas, where Ag is absent. On the other hand, the anodic electrochemical performance correlates well with the catalytic activity for propene oxidation. This suggests that the diffusion of promoting oxygen ions from YSZ via LSCF grains can take place toward Ag nanoparticles and can promote their catalytic activity. The best specific catalytic activity, achieved for a LSCF catalytic layer prepared by screen-printing from the commercial powder, is 800 times higher than that of a pure Ag screen-printed film. This is the first example of Ag nanoparticles electropromoted via the SSEP concept and paves the path to further explore this concept of particular interest from an application point of view. Using SSEP for Ag nanoparticles allows replacing expensive Pt group metals by more abundant Ag. At the same time, the use of the catalytic active phase can be reduced by going from a continuous layer to dispersed self promoted particles.
Round 2
Reviewer 1 Report
My comments
- Answer point 1 is not satisfactory. Please provide e.g. what is the conversion of propene on Ag / LSCF catalysts at higher temperatures, e.g. 400-5000C. Does propene conversion increase at higher temperatures? The 35% conversion is low and does not appear to be sufficient for practical applications.
- The headings in Table 3 (mass m LSCF / mg and mass m Ag / mg) are still not valid. There should be mass of LSCF /mg i mass of Ag / mg.
Author Response
1/ The aim of this study was to investigate the self-sustained electrochemical promotion of Ag nanoparticles dispersed on mixed conducting (O2-/ e-) LSCF layers. Ionic species (oxide ions) can modify the catalytic rate only if the rate determining step is a chemical step (adsorption, desorption, or surface reaction). This chemical regime cannot be achieved at high propene conversions, where the catalytic rate is typically limited by the diffusion. Furthermore, Ag nanoparticles are prone to sinter and coalesce above 300°C. Therefore, we focused the study on catalytic measurements below 300°C. In addition, the technological target is the development of catalytic coatings efficient for indoor air cleaning at low temperatures, typically below 200°C. We agree with the reviewer that the catalytic performances of the Ag/LSCF layers are not yet sufficient but the results validate the role of LSCF in the diffusion of oxygen ions from YSZ toward Ag nanoparticles, leading to self-promoted metallic nanoparticles. This opens new possibilities to further enhance the activity of Ag/LSCF catalytic coatings.
2. The headings in the table have been corrected as suggested.
Round 3
Reviewer 1 Report
I do not understand why it is such a big problem for Authors to check the activity of the Ag /LSCF catalyst at a slightly higher temperature? I can possibly accept the article in this form.